# Epigenetic Studies for Evaluation of NPS Toxicity: Focus on Synthetic Cannabinoids and Cathinones

**DOI:** 10.3390/biomedicines10061398

**Published:** 2022-06-13

**Authors:** Leila Mazdai, Matteo Fabbri, Micaela Tirri, Giorgia Corli, Raffaella Arfè, Beatrice Marchetti, Sabrine Bilel, Eva Bergamin, Rosa Maria Gaudio, Michele Rubini, Fabio De-Giorgio, Matteo Marti

**Affiliations:** 1Department of Neurosciences and Rehabilitation, Section of Medical Biochemistry, Molecular Biology and Genetics, University of Ferrara, 44121 Ferrara, Italy; leila.mazdai@unife.it (L.M.); michele.rubini@unife.it (M.R.); 2Department of Translational Medicine, Section of Legal Medicine, LTTA Center, University of Ferrara, 44121 Ferrara, Italy; fbbmtt1@unife.it (M.F.); micaela.tirri@unife.it (M.T.); giorgia.corli@unife.it (G.C.); raffaella.arfe@unife.it (R.A.); beatrice.marchetti@unife.it (B.M.); bllsrn@unife.it (S.B.); rosamaria.gaudio@unife.it (R.M.G.); 3Section of Legal Medicine, Department of Health Care Surveillance and Bioethics, Università Cattolica del Sacro Cuore, 00168 Rome, Italy; eb95@live.it; 4Fondazione IRCCS Policlinico Universitario A. Gemelli, 00168 Rome, Italy; 5University Center for Studies on Gender Medicine, University of Ferrara, 44121 Ferrara, Italy; 6Department of Anti-Drug Policies, Collaborative Center for the Italian National Early Warning System, Presidency of the Council of Ministers, 11582 Rome, Italy

**Keywords:** new psychoactive substances, pharmacoepigenomics, forensic science

## Abstract

In the recent decade, numerous new psychoactive substances (NPSs) have been added to the illicit drug market. These are synthetized to mimic the effects of classic drugs of abuse (i.e., cannabis, cocaine, etc.), with the purpose of bypassing substance legislations and increasing the pharmacotoxicological effects. To date, research into the acute pharmacological effects of new NPSs is ongoing and necessary in order to provide an appropriate contribution to public health. In fact, multiple examples of NPS-related acute intoxication and mortality have been recorded in the literature. Accordingly, several in vitro and in vivo studies have investigated the pharmacotoxicological profiles of these compounds, revealing that they can cause adverse effects involving various organ systems (i.e., cardiovascular, respiratory effects) and highlighting their potential increased consumption risks. In this sense, NPSs should be regarded as a complex issue that requires continuous monitoring. Moreover, knowledge of long-term NPS effects is lacking. Because genetic and environmental variables may impact NPS responses, epigenetics may aid in understanding the processes behind the harmful events induced by long-term NPS usage. Taken together, “pharmacoepigenomics” may provide a new field of combined study on genetic differences and epigenetic changes in drug reactions that might be predictive in forensic implications.

## 1. Introduction

In the last decade there has been a significant change in the worldwide illicit drugs market. New psychoactive substances (NPSs) have emerged as “legal alternatives” to the well-known addictive and abusive drugs (cannabis, cocaine, heroin and amphetamines) [1]. Otherwise known as “designer drugs”, these novel substances are synthesized in order to maintain and/or increase the pharmacological effect of the classic drugs of abuse but remaining outside the legal controls [2]. Therefore, NPSs represent a public health and regulatory challenge [3]. The number of NPSs monitored and seized by international organizations, such as the European Monitoring Centre for Drugs and Drug Addiction (EMCDDA) and United Nations on Drugs and Crime (UNODC), demonstrate that the NPS market is heterogeneous, rapid and dynamic [2,4]. The NPSs include a wide range of different compounds which can be classified in different categories in order to distinguish their pharmacological effects: stimulant (amphetamines, cathinones, benzofuran, indole and pyrovalerone derivatives), sedatives (synthetic opioids, designed benzodiazepines and Gamma-Hydroxybutyrate), dissociatives (phencyclidine, ketamine, diphenidine and their derivatives), synthetic cannabinoids, and psychedelics (phenethylamines, tryptamines, lysergamides) [2,3,4,5,6,7]. Among these, synthetic cannabinoids and cathinones are most often detected, and are the classes mainly used by young adults (15–35 years old) [8]. In particular, data collected in the European School Survey Project on Alcohol and Other Drugs (ESPAD) reported that students have used synthetic cannabinoids at least once in their lifetime, and it was highlighted that a higher percentage point of males than females used both synthetic cannabinoids and cathinones [9]. Moreover, Bachman and colleagues, analyzing the widespread use of synthetic cannabinoids and cathinones among adolescents in the USA, revealed that there is variability at the interindividual level, but especially between the sexes; in fact, boys are more at risk of using these NPSs than girls [10].

Relying on case reports, NPSs are often involved in polydrug use cases and can induce many and various acute adverse effects [11,12]. Symptoms such as tachycardia, restlessness, agitation, bruxism, seizures, hallucinations, psychosis, unconsciousness, respiratory failures, nausea, vomiting and death have been related to the use of synthetic cathinones [13,14] and cannabinoids [15,16]. In particular, synthetic cathinones have been associated with long-lasting renal failure [17], fetal death [18] and overdose cases typified by symptoms such as stroke, cerebral edema, myocardial infarction and subsequent death [19,20] On the other hand, dysregulation of the endocannabinoid system (ECS) seems to be involved in cardiovascular function [21]. In fact, smoking marijuana has been related to myocardial ischemia, coronary thrombosis and vasospasm [22], and synthetic cannabinoids have shown tachycardia and hypertension effects [23]. Further studies also suggest the proatherogenic role of CB1 receptor [24]. Such evidence may be therefore considered as indicating that these synthetic drugs increase the risk of cardiovascular and cerebrovascular events [25,26], confirmed by the fact that cases of acute ischemic stroke have been reported in young adults after use of synthetic cannabinoids [25]. Biopsy findings have moreover confirmed cases of acute tubular necrosis or acute interstitial nephritis in young patients [27].

In vitro and in vivo studies concerning NPSs were widely present in the literature [28,29,30,31,32,33,34,35,36,37,38]. In particular, Lenzi and colleagues have demonstrated the mutagenic capability of synthetic cathinone and cannabinoids [39,40], underlying their potential toxicity. However, in vitro tests cannot easily predict pharmacokinetics or penetration into the human brain [41,42]. Therefore, animal models have played a key role in preclinical research explaining the neurobiological, psychopathological, behavioral and etiological aspects, and studying drug dependence and acute/chronic abuse [43]. In fact, many in vivo studies show acute effects imputable to NPS-related intoxication [44,45,46,47,48].

Moreover, knowledge of long-term NPS effects is lacking [9,49] and illicit drug responses are complex traits because they may be determined by both genetic and environmental factors. Variations in the expression of drug-response-related proteins (such as drug-metabolizing enzymes (DMEs), drug transporters and therapeutic mechanisms, including drug targets and downstream signal molecules) are other important sources of interindividual variability in drug response [50,51,52,53].

Given this evidence, it may be necessary to understand a possible correlation between the use of NPSs and epigenetic changes. Indeed, data show that epigenetic modifications are linked to changes in development and behavior, but also to genetic disorders and various diseases [54,55]. It is well known that widespread epigenetic changes occur across the entire genome [56]. This potentially reversible “epigenetic” modulation of gene expression occurs through the chemical modification of DNA and histone protein tails [53], or the specific production of regulatory non-coding RNA (ncRNA) [57]. In this regard, it has been observed in patients with myocardial infarction that there are 200 differentially methylated cytosine–phosphate–guanine (CpG) sites in the gene locus involved in cardiac function [58]. In addition, a 100-fold increase was found in the level of circulating miRNAs in the plasma of patients suffering from cardiac events [59,60,61]. Studies concerning cancer prevention and innovative treatments for cancers have led to the label “pharmacoepigenomics” in reference to the emerging field of combined study on genetic variations and epigenetic modifications in drug responses [62,63]. This development may expand the scope of pharmacogenomics and better define the role of each factor involved in variable drug responses.

This narrative review aims to find potential predictive markers of organ damage, considering existing knowledge of NPS toxicology and possibly related epigenetic changes, with a focus on synthetic cathinones and cannabinoids. Our previous study suggested that in vivo synthetic stimulants, such as 4,4′-Dimethylaminorex (4,4′-DMAR), involves physiological, neurobehavioral and neurotoxicological effects, as confirmed by immunohistochemical analysis [64]. In particular, we investigated the effect of 4,4′-DMAR on the expression of specific cerebral cortex markers (oxidative/nitrosative stress, apoptosis and heat shock proteins) commonly known to be indicative of brain damage [65,66]. Therefore, considering that nerve cells require a large number of mitochondria to ensure their normal functioning, and that overproduction of ROS following the administration of stimulants may lead to oxidative damage in the mitochondria and consequently neurotoxicity [67], this could be an excellent starting point to research possible epigenetic markers of NPS-induced physiopathological damage.

### Search Strategy

A systematic search was conducted using Pubmed, considering relevant data from the earliest available date up until January 2022. The following combined search terms were considered: “NPS”, “epigenetics modifications”, “DNA-methylation”, “histone modification”, “non-coding RNA”, “heart damage” and “brain damage”. All duplicates were then removed. Any title that was coherent with this narrative review main topic was included in a subsequent screening based on eligibility of abstract and full text. Studies were considered appropriate if they included the following: (1) animal model (mouse and/or rat); (2) synthetic cannabinoids and cathinones or analogous and traditional drugs of abuse; (3) connection between exposure to the substance and subsequent epigenetic changes. A total of 218 articles were included in the bibliography section (N in vivo, N in vitro). A manual search of the reference list of selected articles was also conducted.

## 2. Epigenetic Changes

In the last two decades, epigenetics was initially defined as “the study of mitotically and/or meiotically heritable changes in gene function that cannot be explained by changes in the DNA sequence” [68,69]. Currently, it is defined as “the study of any phenotypic diversity that is not related to genotypic differences that can be transmitted through cell divisions but are not directly traceable to the DNA sequence” [70,71].

Stretched out linearly, the negatively charged DNA hosted in the nucleus of a mammalian cell is calculated to be approximately 2 m long [72]. In order to compress all this genetic material into ~10 µm diameter of nucleus, the DNA must wrap tightly around positively charged histone proteins to form the nucleosome, the founding unit of the DNA packaging material called chromatin [73]. Chromatin structure ensures access to genetic information even in the presence of a highly protective environment, which under physiological conditions makes DNA essentially inaccessible. Nevertheless, evolution has developed enzymatic systems that make it possible to modulate chromatin, and consequently nucleosomes, to ensure “regulated” access to DNA by competent systems [74].

Tightly packed DNA is considered heterochromatin, while loosely packed DNA is considered euchromatin [75]. As opposed to euchromatin, which is freely accessible to the transcriptional machinery and actively transcribed, heterochromatin is generally transcriptionally inactive, although a recent study investigated its potential activity [76]. Changes in the components that make up chromatin (Table 1) itself make it more or less accessible to transcription, constituting epigenetic inheritance (Figure 1) [77].

### 2.1. Histones Modification

Post-translational modifications on the residues of the histone tails of nucleosomes allow the relaxing of chromatin and its condensation, which enables regulation of gene expression, DNA replication, recombination and repair processes [62]. The first identified histone modifications were acetylation and methylation, followed by ubiquitination and phosphorylation [78]. All histone modifications (which are directed by enzymatic reactions) exert their action by means of two mechanisms: through a local or remote alteration of the whole chromatin structure, and through a positive or negative modification of the binding to effector molecules [79]. The functional consequences of histone modifications occur through “reader” proteins that bind to specific modified residues and make transcriptional changes. The enzymes (acetyltransferases (HAT) catalyse acetylation and deacetylases (HDAC) deacetylation, while methyltransferases (HMT) catalyze methylation and demethylases (HDM) demethylation])involved in these various covalent histone modifications can be labeled as “writers” and “erasers” [79]. Among this, the best-known epigenetic mechanism of histone alteration in the brain is the post-translational covalent modification of the N-terminal tails of histones in different amino acid residues [80].

The methods used to study histone modifications are mainly based on systems using monoclonal and polyclonal antibodies for target recognition, such as chromatin immunoprecipitation [81] and proteomic methods, have only recently been developed [74].

### 2.2. DNA Methylation

DNA methylation is a process that induces gene silencing and heterochromatin formation [82]. It is involved in the regulation of gene expression, genomic imprinting, X-chromosome inactivation and silencing of centromeric regions [83,84].

The most common DNA modification is the covalent transfer of the methyl group (-CH_3_) to cytosine located within the CpG island, long over 500 base pairs (bp) [85,86]. This produces 5-methylCytosine (5-mC) in the gene promoter region [87,88]. The level of methylation of the CpG islands of the regulatory region of a gene is associated with the transcription levels of that gene [89]. Hypomethylation typically enhances gene expression [90], while hypermethylation is associated with gene silencing (as opposed to histone methylation) [91]. There is of course also genetic variability between individuals in the density of CpG sites, which influences potential methylation levels and thus affects the regulatory activities of reference genes [91].

Compared with readily reversible histone tail modifications, DNA methylation is considered a more stable epigenetic change, altering the chromatin structure from an opened—transcriptionally active—to closed—transcriptionally inactive—state [92].

Historically, methods for studying DNA methylation can be specific or non-specific. In particular, specific methods provided a global assessment of methylation levels in cells. On the other hand, non-specific methods make a qualitative assessment of changes occurring in a given DNA segment in different cells after treatment with bioactive molecules [93]. Such detection methods are based on three principles: DNA digestion with restriction enzymes, use of anti-methylcytosine antibodies or methyl-binding domain proteins to enrich methylated genomic DNA fragments and DNA conversion by bisulfite treatment and sequencing [94].

In recent years, DNA methylation analysis has been increasingly available for their potential use in biomedical research, and has been performed through genome-wide and high-throughput methods [84]. These advances have significantly accelerated epigenomic research and opened up new perspectives, in particular through the development and application of massively parallel sequencing technologies [95].

### 2.3. Non-Coding RNA

Non-coding RNAs (ncRNAs) have been defined as RNAs having information and functions but not having the ability to encode proteins [96], and they have been shown to play important roles in the regulation of cellular functions [97]. NcRNAs are epigenetic markers for prognosis, diagnosis and treatment detectable in both tissues and biological fluids that can influence gene expression [98]. Generally, ncRNAs can be divided into housekeeping or regulatory types [99]. Among the housekeeping transfer RNA (tRNA), rRNA and small nuclear RNA (snRNA) have been identified. Regulatory ncRNAs include various RNA types distinguishable in long non-coding RNA (lncRNA) and small non-coding RNA (sncRNAs) [100]. LncRNAs have been placed into six different categories relative to their proximity to the protein-coding genes, and categorized as sense or antisense [99]. Moreover, an increasing number of lncRNAs have been shown to have crucial roles in several biological processes, such as X-chromosome inactivation and imprinting [101,102,103], maintenance of nuclear architecture [104], epigenetic control of gene expression and promoter-specific gene regulation [105]. On the other hand, sncRNAs include microRNA (miRNA), P-element-induced Wimpy testis (PIWI)-interacting RNAs (piRNAs) and short interfering RNA (siRNA) [106]. Furthermore, circular RNA (circRNA) has been labeled as a special class of non-coding RNA [107].

The analysis techniques for these sequences are similar to those conventionally used to study other RNAs. They are techniques based on quantitative PCR, sequencing and bioinformatic analysis [108].

## 3. Addiction and Epigenetic Mechanisms Related to Drugs of Abuse

Addiction is described as compulsive use, seeking and craving of drugs, and seems to be related to pathological neurobiological changes in neural processes that normally serve reward-related learning [109].

Furthermore, early life experiences and environmental factors together with genetic susceptibility result in addiction development [110]. Drug abuse is characterized by two different stages [111]. In the initial period, after an occasional intake of drugs, the dopaminergic system in NAc triggers pleasurable feelings that results in the uncontrolled pursuit and use of such substances [112]. In the second stage, different clinical processes occur, which are related to more complex and long-lasting mechanisms implicating alterations in other signal neurotransmitters [113] (e.g., outcomes on glutamate receptors after exposure to methamphetamine; Table 2 [114]). Recent studies suggest that different drugs of abuse induce epigenetic changes [115,116,117] in gene expression and post-transcriptional regulation (see Table 2 [118,119,120,121]). Indeed, the role of microRNAs in drug addiction and neuroplasticity [115,116,117] due to their ability to regulate gene expression has been widely demonstrated [122,123]. This suggests that different epigenetic changes are potentially related to drug addiction [124,125,126].

Over the years, attempts have been made to define the mechanisms by which exposure to a drug of abuse alters mRNA levels through chromatin regulation mechanisms or by activating/inactivating additional genes for altered expression after a period of abstinence [127].

However, previous studies have pointed out that traditional drugs of abuse, such as cannabis (Table 2 [128]) and cocaine (Table 3 [129]), may influence gene expression by inducing epigenetic changes. Indeed, the role of the endocannabinoid system (ES) in reward-related learning and addiction was already investigated [130,131]. Specifically, it has been shown that the use of cannabis may interfere with endocannabinoid signalling and affect the neural pathways that regulate reward-related learning and cognition [131], thus strengthening the hypothesis that synthetic cannabinoid may affect the same neural pathways. Therefore, the ES is possibly sensitive to environmental epigenetic cues. In fact, there is evidence of genetically related variations in reactivity to stress and negative emotional states, and drug craving as a result of an altered endocannabinoid system feature [132]. In particular, the endocannabinoid system undergoes dynamic changes during adolescence, which is when anxiety disorders often emerge [132]. This could possibly explain data showing that cannabis use among adolescents has been associated with an increased risk of subsequent cocaine intake [133], and abuse of synthetic cannabinoids has been associated with multiple drug use, including stimulants [134], and induces a behavioral change in exposed animals (see Table 3, [135]). To confirm this, a recent study has shown that cannabinoid exposure in adolescent can change the behavioral and epigenetic response to cocaine in rodents [136,137]. In addition, attention was paid to the proenkephalin opioid neuropeptide (Penk) gene in NAc that directly regulates heroin addiction. Adolescent rats exposed to THC overexpressed Penk and reported reduced methylation of histone H3 lysine 9 in NAc (Table 3, [138]).

Furthermore, studies on exposure to psychostimulants, such as cocaine, have revealed the occurrence of epigenetic changes previously attributed to psychiatric disorders (drug addiction, depression or Rett syndrome) since the 2000s [61,62,139]; (Table 2, [140,141]). In particular, chronic cocaine exposure has been shown to recruit histone acetyltransferases and regulate histone acetylation/deacetylation in NAc [142]; (Table 2, [114,115,116,117,118,119,120,121,122,123,124,125,126,127,128,129,130,131,132,133,134,135,136,137,138,139,140,141,142,143]). The same study has shown that the main effect of chronic cocaine exposure is gene activation, with more genes showing H3 or H4 hyperacetylation than hypoacetylation (Table 2, [143]). Moreover, changes at the epigenetic level occur in mice repeatedly administered with 3,4-methylenedioxymethamphetamine (MDMA), and this could be related to the cardiotoxicity induced by such drugs of abuse [144]. The knowledge obtained in the context of abuse (e.g., use of methionine as a therapeutic target for cocaine addiction (Table 2, [145,146]) and pathological information could be used to investigate markers that define therapeutic targets or intoxication antidotes to avoid permanent damage to the organism.

## 4. Overview of Epigenetic Factors and Their Clinical-Toxicological Use

Recent studies have shown that ageing is a complex multifactorial mechanism shared by all living organisms and is expressed by the gradual decline of physiological functions [155] and influenced by various genetic, lifestyle and environmental factors [156]. Therefore, it is important to differentiate chronological age from biological age [157]. Individuals of similar chronological age may show very different susceptibilities to age-related diseases and death, which presumably reflects differences in biological ageing processes [155]. In particular, DNA methylation patterns change in the process of aging and contribute to age-related disease development [158]. Diverse epigenetic changes occur during the lifetimes of mammals [159]. In particular, studies have shown a correlation between age and DNA methylation at the level of single CpG sites [160,161]. On this basis, predictive models have been built in order to quantify age-related phenotypes or outcomes, such as diseases (cardiovascular and neurodegenerative conditions) or mortality [161,162]; such models are labelled as “epigenetic clocks”. Epigenetic age deceleration has been associated with longevity [163], strengthening the above-mentioned assumption. Taken together, these notions suggest that chronic use of certain drugs could lead to neurotoxicity effects proportionate to increasing subject age—being amplified in adult and elderly animals compared to young ones [164,165,166]. In this context, differences have been observed in terms of epigenetic modifications that interact differentially in adult and adolescent subjects under the same conditions of exposure to substances of abuse (Table 3, [148]).

Furthermore, a possible gender-related difference among such epigenetic changes has been studied. Specifically, a recent study has shown sex-specific variations in the DNA methylation patterns of two distinct genes (*FIGN* and *PRR4*) [167]. These sex disparities have been previously confirmed by studies on the mouse hippocampus and the human frontal cortex [168]. According to Global Health Observatory (GHO) data, nowadays global life expectancy at birth is 76 years for females and 71 years for males [169]. In agreement with this evidence, data have shown that male’ mortality rates due to cardiovascular, cancer and Parkinson’s disease are higher than those of females at a given age. On the other hand, females show an increased risk of Alzheimer’s and autoimmune diseases [167].

Furthermore, gender can influence subjective effects and pharmacotoxicological responses to drugs [170] and can also present differences in the activity of specific enzymes involved in drug metabolism [171]. In particular, CYP450 family enzymes play a crucial role [172]. The isoenzymes CYP2D6, CYP3A4, CYP1A1, CYP1A2 and CYP2C19 are involved in the metabolism of most psychiatric drugs and many other drugs commonly used and prescribed in daily practice (beta-blockers, opioid analgesics, anticonvulsants, antihistamines, cortisones and the macrolide antibiotics) [172,173]. Studies have also confirmed the involvement of the above-mentioned enzymes in the metabolism of synthetic cannabinoids [174,175] and cathinones [176]. For example, the cytochrome P450 3A4 is abundantly expressed in the liver, but its activity is higher in women than in men [177]. This could be due to female-specific issues, such as pregnancy, menopause, oral contraceptive use and menstruation, suggesting a possible role played by sexual hormone levels [177]. Other features, such as body weight and the amount of adipose tissue, depending on the subjects themselves, should be considered [171].

Compelling studies have shown that NPSs, such as synthetic cannabinoids and cathinone, can induce gender-related effects on animals [170]. In certain respects, the effects of 3,4-Methylenedioxypyrovalerone (MDPV) on cardiovascular parameters are deeper and long-lasting in males than females [178]. However, Fattore and colleagues have reported that females are more susceptible to the reward-seeking and anxiety effects respectively induced by synthetic cannabinoids and cathinones than males [179]. Different studies have also investigated effects of prenatal exposure to cocaine [147,180,181,182,183,184]. Specifically, it has been demonstrated that cocaine can induce sex-dependent epigenetic changes during the gestational period [180] and may increase the heart’s vulnerability to ischemic damage during adulthood [183]. Moreover, mice prenatal cocaine exposure has been shown to be related to alteration of the genes involved in Wnt and the cadherin system [184]. This suggests that prenatal exposure to cocaine leads to an increased susceptibility of the heart to ischaemic damage in the adult offspring (F1), due to decreased *PKC* gene expression. Thus, programming of *PKC* gene expression patterns in the heart already occurs in utero (F0) (Table 2, [147]).

These considerations reflect the danger of synthetic cannabinoids, precisely because Δ^9^-THC can cross the placental barrier and come into contact with the fetus [185]. This can cause defective development of the child’s brain and produce neurobehavioral toxicity [186]. In addition, studies on the effects of synthetic cannabinoids such as JWH-018 report neurobehavioral alterations and in particular evidence of substance dependence [187].

### Epigenetic Inheritance

Multigenerational epigenetic inheritance is defined as the germline-mediated heritage of epigenetic information between generations after direct environmental influences, which results in phenotypic variation in the offspring [188,189]. However, some literature considers transgenerational epigenetic inheritance in the absence of continued direct environmental influences [190]. Specifically, exposure to a variety of environmental factors in F_0_ of both genders before gestation may have a direct impact on the germ cells, which will affect the F_1_ generation. Therefore, the phenotypes found in F_0_ and F_1_ animals are considered multigenerational. In contrast, only those traits that persist in the F_2_ generation and beyond are considered examples of transgenerational inheritance (Figure 2) [190]. This phenomenon has been investigated in detail with regard to X-chromosome inactivation [191] and exposure to drugs of abuse [27]. Drug exposure during embryonic development simultaneously exposes the developing fetus (F_1_) and germ cells (F_2_) to the effects of the drug. Following the same pattern, drug-using parents expose their germ cells and consequently the F_1_ generation [192]. Furthermore, it was found that perinatal exposure to CBD induced changes in DNA methylation levels in F_1_ generation associated with increased anxiety and improved memory behavior in a sex-specific manner [193]. These changes have been associated with exposure to environmental factors that could also induce epigenetic changes in sperm or ova (Table 3 [125,191,192]). Previous studies have shown that prenatal THC exposure can alter cognitive function, emotional reactivity [151] and responses to drugs of abuse, such as methamphetamine [194] and opiates (Table 3, [193]). Direct exposure to drugs of abuse in the embryo (F_3_) and/or in the parents (F_2_) has been also proven to cause transgenerationally inherited alterations in DNA methylation (Figure 2) [195]. In support of this, the NAc epigenome was analyzed in a Long Evans rat model, identifying 1027 differentially methylated regions associated with parental exposure to THC in F1 adults (Table 3, [152]). These studies provide new information on drug-related intergenerational epigenetic effects and are a starting point for the study of neurobiological effects underlying drug abuse vulnerability.

Concerning the use of synthetic cannabinoids, scientific evidence shows that there is a predisposition to neurodevelopmental disorders (i.e., schizophrenia and autism spectrum disorders) in the offspring of women exposed to this class of substance during pregnancy [196]; (Table 3, [135,153,154]). Moreover, a recent study has connected cannabis exposure during the perinatal period to epigenetic changes in animal models [197]. Epigenetic changes possibly induced by NPS abuse have not yet been investigated. Despite this, recent study has revealed that THJ-2201 and 5F-PB22 affected neuronal differentiation, and this could result in neurodevelopmental disorders if synthetic cannabinoids use occurs before or during pregnancy [198].

## 5. New Perspectives in the Forensic Field

From the perspective of forensic medicine, several reports concerning the effects of NPSs can be found in the literature [32,199]. Indeed, there is no doubt that the availability and consumption of these substances has become increasingly important in recent decades, with the number of deaths attributed to NPSs growing year by year [200].

However, detailed data on the number of substances marketed and consumed, and on the extent and type of organ damage caused by them, are still insufficient [201]. The advent of NPSs has called into question traditional methods of drug detection, monitoring, surveillance, and control [200]; for instance, in the clinical setting, rapid immunoassay screening tests typically cannot detect NPSs [202]. Similarly, in the domain of forensic toxicology, NPSs may be difficult to distinguish from more well-known illicit substances and other psychotropic chemicals due to a lack of available data on their pharmacology, toxicology, and health impacts [203]. Indeed, NPS research is typically overlooked in regular screens, and traditional approaches might not be as effective for detecting NPSs [200]. Clinicians are thus required to be familiar with the many classes of these compounds, as well as their effects, in order to apply this information to the specific case and carry out focused toxicological assessments [204]. Given the numerous challenges associated with NPS detection and characterization, as well as the relative lack of pertinent data in the literature, it is obvious that identifying as many methods as possible to obtain information on these specific substances of abuse is becoming increasingly important. Forensic epigenetics is a relatively new field of research with a wide range of potential applications. In forensic medicine, DNA methylation is typically chosen over other epigenetic alterations because of its in vitro stability and great sensitivity in terms of the amount of DNA required [205]. Based on the notion that environmental stimuli can cause individual epigenomic variations [205,206], forensic investigations apply DNA methylation to the identification of tissues and cells [207], to the determination of both sex and age of individuals [208,209], and to the differentiation between monozygotic twins [210]. Furthermore, DNA methylation can disclose information about a person’s socioeconomic status, diet, physical activity, alcohol use [211], smoking status and drug use [212], and thus may aid in the discovery of specific markers that can provide more detailed information about the characteristics and effects of NPSs.

Hence, in order to identify various effective methods of prevention and treatment, future preclinical, clinical and forensic research on synthetic cannabinoids and cathinones could benefit from the support of epigenetics. Moreover, given the complexity of the NPS issue, there is no doubt that combining different methods of investigation with integrated pharmacological and psychological strategies aimed at treating intoxication symptoms will yield significant benefits. Potential investigation methods for this purpose include forensic investigation, pre-clinical in vitro research and in vivo models of mice and other animal species to provide important tools for analyzing human drug metabolism, as well as clinical-level investigations. In addition, the modification of treatment regimens for the more well-known drugs of abuse can help address the rising incidence of intoxication reported following the use of synthetic cathinones and cannabinoids.

## 6. Conclusions

This review aims to provide an overview of epigenetic modulation and possible hyper- or hypo-expression of epigenetic markers of tissue damage. In fact, the epigenetic structure can be studied in animal models throughout the life cycle of an organism, from conception to adulthood and old age [213]. An ideal model needs to be accessible at all stages of prenatal and postnatal life, and to have a reasonable time span to enable assessment of all possible outcomes of an experiment [214]. Thus, animal models allow the investigation of the probable interplay between epigenetic changes and effects caused by the intake of synthetic cannabinoids and cathinone, focusing on gender related differences. Given the great potential toxicological and forensic value of understanding epigenetic changes induced by exposure to drugs of abuse, the overall strength of the present narrative review is the suggestion of a translational evaluation of the pharmacotoxicological effects of NPSs widely reported by preclinical and clinical literature. However, this aspect also represents a weakness, because of the great variety of environmental and non-environmental factors that can influence epigenomic changes.

## Figures and Tables

**Figure 1 biomedicines-10-01398-f001:**
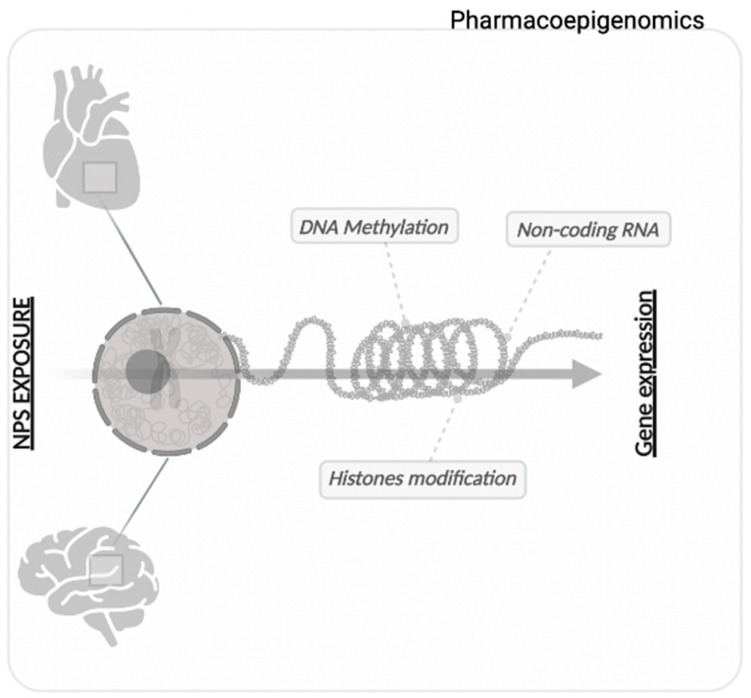
Schematic illustration of the epigenetic changes that are of toxicological relevance for finding useful markers in a context of predicting cardiac and brain damage. Exposure to an NPS induces changes in the epigenetic landscape that could provide the basis for a panel of pharmacoepigenetic markers to predict toxicological damage to vital organs. The study of gene expression variations in a pharmacoepigenomic context is indicative of the variability between individuals in their responses to drugs or drugs of abuse.

**Figure 2 biomedicines-10-01398-f002:**
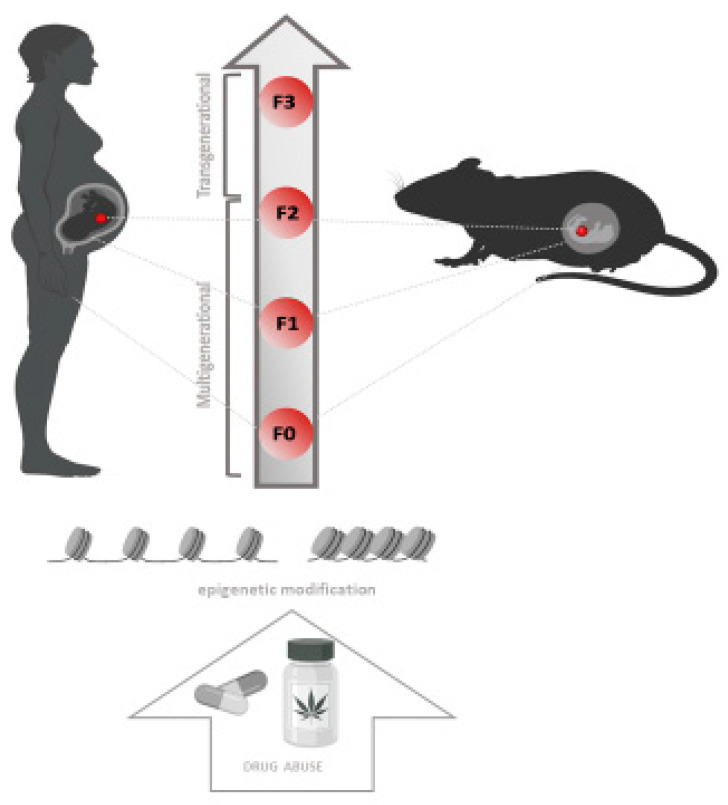
Epigenetic inheritance.

**Table 1 biomedicines-10-01398-t001:** Summary of all epigenetic changes.

Epigenetic Mechanisms	Epigenetic Modification	Effects on Gene Expression
Histone modification	Addition of an acetyl group (Ac) to the amine group of the lysine residues of histones H2B, H3 and H4; addition of one or more methyl groups (Me) on lysine or arginine residues preferentially found on histone tails.	Activation or repression of gene transcription.
DNA methylation	Addition of a methyl (Me) group that occurs preferentially in C- and G-rich genomic region CpG islands.	Gene repression.
Non-coding RNA	Different RNAs—not translated into proteins—that influence gene expression at the transcriptional and translational level.	Regulation of gene expression (miRNAs binds mRNAs in the 3’UTR region, which promotes their degradation or prevents their translation in protein).

**Table 2 biomedicines-10-01398-t002:** Summary of information obtained from the available literature about the main epigenetic changes observed in rodents after intake of traditional stimulants. We report: the substances, the effects found after epigenetic modification, information on the animal model used (by specifying genotype, gender and age), and which detection method is the most appropriate to identify the previously highlighted epigenetic modification.

Substance	Target	Epigenetic Modification	Effect	Animal Information			Tissue/Cell Type	DNA Methylation Method	References
				Genotype	Gender	Age			
Methamphetamine	*GluA1*	Hypoacetylation H4	Downregulation	Sprague–Dawley rats	Male	Adult	Striatum	Immunoblot/ChIP/Antibody	[114]
	*GluA2*								
Cocaine	*fosB*	Increased CBP acetylation H4	Upregulation	C57BL/6J	Crossed C57BL6J mutant males with BALBc females to generate the F1 hybrids	Adult	Striatum	ChIP/qPCR/Immunoblotting	[118]
				BALBc mice					
Cocaine	*BDNF*	Increased	Upregulation	Sprague–Dawley rats	Male	Adult 10–14 weeks old	Striatum	ChIP/qPCR/RT-PCR/immunohistochemistry	[119]
	*Cdk5*	Acetylation H4		Bl6/C57 mice					
Methamphetamine	5 of *IEGs*	Increased DNA methylation	Downregulation	CD-1 mice	Male	Adult	PFC	RT2 Profiler PCR/qPCR/Pyrosequencing	[120]
	*Grm1*						Hippocampus		
Methamphetamine	*FosB*	Increased acetylation H4	Upregulation	C57BL/6 mice	Male	Adult	Striatum	ChIP/Western Blot	[121]
		Increase CREB phosphorylation							
Cocaine	*N/A*	DNMT3A	Downregulation	C57BL/6J mice and Long Evans rats	Male	Adult	NAc	ChIP promoter analysis/Global DNA methylation analysis	[126]
		Increased DNA methylation							
Cocaine	55 genes	TET1	Downregulation Alternative splicing	C57BL/6J mice	Male	Adult 8–10 weeks old	NAc	qPCR/western blotting/immunohistochemistry/stereotaxic viral manipulations/ChIP-seq/RNA-seq	[129]
		Increased DNA methylation							
Cocaine	*PP1c*	Increased DNA methylation DNMT3A and DNMT3B	Downregulation and Upregulation of FosB	Mice C57BL/6	Male	Adult	NAc	qPCR/ChIP/MeDIP/western Blotting	[140]
Cocaine	*Cdkl5*	Increased DNA methylation (MeCP2)	Downregulation	Wistar rats	Male	Adult 8–9 weeks old	NAc	RT-PCR—qPCR/Bisulfite sequencing/Chromatin immunoprecipitation/Immunohistochemistry	[141]
Cocaine	*cFos, FosB, BDNF* e *Cdk5*	Increased Acetylation H3 and H4	Upregulation	Sprague–Dawley rats	Male	Adult	Striatum	ChIP	[119]
Cocaine	Δ*Φοσ*	Increased Acetylation H3 and H4	Upregulation: *sirt1 sirt2*	C57BL/6 mice	Male	10–12 weeks	NAc	ChIP/Array	[143]
	*CREB*								
Cocaine	172 genes	Decrease Histone Deacetylation HDAC5	Upregulation	Mice C57BL/6	N/A	Not adult	NAc	Western Blotting/Immunohistochemistry/ChIP/qPCR/RT-PCR/Microarrays	[142]
Cocaine	*N/A*	DNMT3B	Upregulation	C57/BL6 mice	Male	Adult	PFC	Global DNA methylation analysis, qPCR, Western blotting	[145]
		Decrease DNA methylation							
Cocaine	57 genes	DNMT3A Increased DNA methylation	Up-/downregulation	C57BL/6 mice	Male	Adult aged 5–6 months	NAc	MicroArray Illumina/qPCR/MeDIP	[146]
Cocaine	*PKCε*	Increased DNA methylation	Downregulation	Sprague–Dawley rats	Pregnant	Fetal (21 days)	Heart	Quantitative Methylation-Specific PCR/ChIP	[147]

**Table 3 biomedicines-10-01398-t003:** Summarize of information obtained from the available literature about the main epigenetic changes observed in rodents after intake of traditional and new synthetic cannabinoids. We reported: the substances, the effects found after epigenetic modification, information on the animal model used (by specifying genotype, gender and age), and which detection method is the most appropriate to identify the previously highlighted epigenetic modification.

Substance	Target	Epigenetic Modification	Effect	Animal Information			Tissue/Cell Type	DNA Methylation Method	References
				Genotype	Gender	Age			
Δ⁹--THC	*Appbp2 CD27*	LncRNA	Downregulation of miR-17/92 cluster and miR-374b/421 cluster Up-regulation of miR-146	C57BL/6J mice	Female	6–7 weeks old	Lymph node cells CD4+	RNA-seq	[128]
WIN55212.2	*Rgs7*	Hypermethylation	Downregulation	C57Bl6/J mice	Male	Adolescent aged 4 weeks	Hippocampal CA regions	qPCR/RT--PCR	[135]
Δ⁹--THC	*Penk*	Decrease H3K9 Methylation	Upregulation	Long Evans rats	Male	Adolescents 21-day-old	NAc	ChIP	[138]
Δ⁹--THC	N/A	H3K9me2 H3K9me3 H3K9ac H3K14ac	Both	Sprague–Dawley rats	Female	Adolescent (35–45 postnatal day) and adult (75–85 postnatal day)	Hippo NAc Amy	Western Blot	[148]
Δ⁹--THC	177 genes	DNA Methylation	Both	Sprague–Dawley rats	Male	Nine-week-old, sexually mature	Semen	Bisulfite Sequencing/Pyrosequencing	[149]
JWH133	*CB2*	Upregulation of H3K4me3 Downregulating H3K9me2	Upregulation Prdm9 c-Kit Stra8	Swiss CD-1 mice	Male	Seven-day-old	SPG cells (Spermatogonia)	qPCR/RT--PCR/ChIP/Western Blot	[150]
Δ⁹--THC	*Drd2*	Increased 2meH3K9	Decrease 3meH3K4	Long Evans rat	Male and female	Adult	NAc	In Situ Hybridization Histochemistr/ChIP/	[151]
			and mRNA level						
Δ⁹--THC	1027	DNA Methylation	Alterated	Long Evan rats	Male and female	Adolescent	NAc	Enhanced Reduced Representation Bisulfite Sequencing	[152]
	genes								
Δ⁹--THC	Genes associated with plasticity	Increase Suv39H1 and H3K9me3	Downregulation	Sprague-Dawley rats	Female	Adolescent and adult	PFC	RT2 Profile PCR Array/ChIP	[153]
HU-210	Dlk1-Dio3-imprinted domain	Differential miRNA expression	Both	Wistar rats	Female Male	Pregnant Offspring	Left and right hemispheres of the entorhinal cortex (EC)	qPCR/RT--PCR	[154]

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
