# Peer review of "Epigenetic Studies for Evaluation of NPS Toxicity: Focus on Synthetic Cannabinoids and Cathinones"

_biomedicines, 2022, doi:10.3390/biomedicines10061398_

Round 1

Reviewer 1 Report

Thank you for giving me the opportunity to review this manuscript.

This manuscript reviews “pharmacoepigenomics" and its relation to substance use. Especially, it explains molecular mechanism of substance use on epigenomics.

I think this manuscript is well written and very interesting.

It is better to revise the manuscript based on the points below.

1) What about the future perspectives in clinical research (as well as non-clinical research) on how to prevent and treat synthetic cannabinoids and cathinones?? If the authors reviewed these aspects, it may be more interesting to us.

2) In page 7, "there is evidence of genetic related variation of reactivity to stress and negative emotional states, and drug craving as a result of altered endocannabinoid system feature." These are interesting to mental health professionals. If possible, please explain how to control for reactivity to stress and negative emotional states by preventing/treating endocannabinoid system feature.

3) In Figure 1, Please enlarge the words of "DND methylation", "Non-coding RNA", and "histones modification" to make them more visible.

I think it is better to revise the manuscript.

Author Response

We thank the Reviewer 1 for his/her evaluation of our manuscript and for helpful concerns to improve the review. In this revised version of the work, we have addressed the major concerns of the referee (highlighted in yellow).

Rev1Q1: What about the future perspectives in clinical research (as well as non-clinical research) on how to prevent and treat synthetic cannabinoids and cathinones?? If the authors reviewed these aspects, it may be more interesting to us.

AA: We agree with Reviewer 1 and added a short paragraph in the section “New perspectives in the forensic field” that went to briefly analyze the future perspectives in terms of prevention and treatment of synthetic cannabinoids and cathinones:

Hence, in order to identify various effective methods of prevention and treatment, future preclinical, clinical and forensic research on synthetic cannabinoids and cathinones could benefit from the support of epigenetics. Moreover, given the complexity of the NPS issue, there is no doubt that combining different methods of investigations, such as forensic investigations, pre-clinical research with in vitro, as well as in vivo models of mice and other animal species, which provide important tools for analyzing human drug metabolism, and clinical-level investigations, with combined and integrated pharmacological and psychological strategies aimed at treating intoxication symptoms, will yield significant benefits. In addition, the modification of treatment regimens for the more well-known drugs of abuse can help address the rising incidence of intoxications reported following the use of synthetic cathinones and cannabinoids.”

Rev1Q2: In page 7, "there is evidence of genetic related variation of reactivity to stress and negative emotional states, and drug craving as a result of altered endocannabinoid system feature." These are interesting to mental health professionals. If possible, please explain how to control for reactivity to stress and negative emotional states by preventing/treating endocannabinoid system feature.

AA: We agree with Reviewer 1 about the scientific value of evaluating endocannabinoid system features related to reactivity to stress and negative emotional states or possible ways to control or treat them. However, given the heterogeneity of numerous pathways involved in these main mental issues, we did not investigate this point in detail to avoid a defective overview and straying of the attention from the main focus of this review. Therefore, we attempt to briefly clarify this point as follows.

Due to neurogenesis hypothesis of depression disorder development, the key role played by endocannabinoids as well as CB1 and CB2 receptors has been previously investigated. In particular, CB1 receptors expression in development brain regions is suggestive of the involvement of their activation in promoting neurogenesis (Buckely et al., 1998; Williams et al., 2003; Berghuis et al., 2007). Furthermore, further studies interestingly pointed out that CB2 overexpression inhibited depressive-like behaviors and prevented changes in BDNF gene and protein expression under stress in transgenic mice (Garcia-Gutiérrez et al., 2010). Thus, it could explain the potential epigenetic mechanisms behind responses to stress and negative emotional states, pointing out a possible prevention/treatment target.

  1. Buckley NE, Hansson S, Harta G, Mezey E. 1998. Expression of the CB1 and CB2 receptor messenger RNAs during embryonic development in the rat. Neuroscience 82:1131–49

  2. Williams EJ, Walsh FS, Doherty P. 2003. The FGF receptor uses the endocannabinoid signaling system to couple to an axonal growth response. J.Cell Biol. 160:481–86

  3. Berghuis P, Rajnicek AM, Morozov YM, Ross R, Mulder J. 2007. Hardwiring the brain: Endocannabinoids shape neuronal connectivity. Science 316:1212–16

  4. García-Gutierrez MS, Pérez-Ortiz JM, Gutiérrez-Adán A, Manzanares J. 2010. Depression-resistant en-endophenotype in mice overexpressing cannabinoid CB(2) receptors. Br. J. Pharmacol. 160:1773–84

Rev1Q3: In Figure 1, Please enlarge the words of "DND methylation", "Non-coding RNA", and "histones modification" to make them more visible.

AA: As suggested by Reviewer 1 we enlarged above mention words and replaced the Figure 1 with the new version.

Reviewer 2 Report

The present study is a narrative review on the effects of New Psychoactive Substances on epigenetic changes. The paper is well-written and of interest for the journal. However, several minor changes should be made before publishing it.

In the abstract section the authors are describing that they included information based on in vitro and in vivo studies investigating the pharmaco-toxicological profile. However, the authors did not explain the methods they used to select the studies. I would recommend to add a brief paragraph to clarify how the papers were selected. How many papers were included? By design: in vitro, in vivo?

At the end of the introduction section, the authors are reporting the main aims of this work. It should be clarified that this is a narrative review.

After the aims section, I recommend to add a brief paragraph reporting the inclusion/exclusion criteria, how did they select the study, how are the authors dividing the review.

In the section "Epigenetic changes" the authors are dividing them into: Histones metilation, DNA Methylation and non-coding RNA. A table summarizing these three parts would be helpful. Figure 1 is also useful.

There is no apparent discussion section at the end of the manuscript. The authors are finalizing the paper with a "New perspectives in the forensic field" and "conclusions" section.

I would add a brief discussion section based on the comparison of epigenetic changes induced by the use of these NPS, and those induced by any other drugs. Why is important to consider the effects of NPS toxicity?

A Limitations and strenghts section is also needed.

Author Response

Response to Reviewer 2

We thank the Reviewer 2 for his/her evaluation of our manuscript and for helpful concerns to improve the article. In this revised version of the work we have addressed the major concerns of the referee (highlighted in blue).

Rev2Q1: In the abstract section the authors are describing that they included information based on in vitro and in vivo studies investigating the pharmaco-toxicological profile. However, the authors did not explain the methods they used to select the studies. I would recommend to add a brief paragraph to clarify how the papers were selected. How many papers were included? By design: in vitro, in vivo?

AA: Being NPS and epigenetic modifications an unresearched topic, the literature on NPS and epigenetic modifications is particularly limited. The literature search was carried out by including and evaluating the following specifications:

- studies on an animal model and in particular on a rodent model (mouse and/or rat);

- studies concerning the two classes of substances under consideration: synthetic cannabinoids and synthetic cathinones, using references for analogous and traditional drugs of abuse (e.g. stimulants for synthetic cathinones);

- studies highlighting the connection between exposure to the substance and subsequent epigenetic changes impacting on the body's health.

Moreover, we agree with Reviewer 2 and add a brief paragraph to clarify how and how many papers were selected as follows:

Search strategy

A systematic search was conducted using Pubmed, considering the erliest available datas until January 2022. The following combined search terms were considered: “NPS”, “epigenetics modifications”, “DNA-methylation”, “histone modification”, “non-coding RNA”, “heart damage” and “brain damage”. All duplicates were then removed. Any title that was coherent with this narrative review main topic were included in a subsequent screaning based on eligibility of abstract and full text. Studies were considered appropriate by including and evaluating the following specifications: (1) animal model (mouse and/or rat); (2) synthetic cannabinoids and cathinones or analogous and traditional drugs of abuse; (3) connection between exposure to the substance and subsequent epigenetic changes. A total numbers of 218 articles were included in bibliography section (N in vivo, N in vitro). A manual search of the references list of selected articles was also conducted.”

Rev2Q2: At the end of the introduction section, the authors are reporting the main aims of this work. It should be clarified that this is a narrative review

AA: We agree with the Reviewer 2 and modified the sentence: “This narrative review aims to find..”

Rev2Q3: After the aims section, I recommend to add a brief paragraph reporting the inclusion/exclusion criteria, how did they select the study, how are the authors dividing the review

AA: We agree with the Reviewer 2 and add the paragraph. See answer to Rev2Q1.

Rev2Q4: In the section "Epigenetic changes" the authors are dividing them into: Histones metilation, DNA Methylation and non-coding RNA. A table summarizing these three parts would be helpful. Figure 1 is also useful.

AA: We agree with Reviewer 2 and added a table summarizing the epigenetic changes mechanisms.

Rev2Q5: There is no apparent discussion section at the end of the manuscript. The authors are finalizing the paper with a "New perspectives in the forensic field" and "conclusions" section. I would add a brief discussion section based on the comparison of epigenetic changes induced by the use of these NPS, and those induced by any other drugs. Why is important to consider the effects of NPS toxicity?

AA: Due to the fact that this is a narrative review, we prefer to conclude with New perspectives in the forensic field highlighting the great availability of different NPS in the illicit market and the concurrent lack of information about them in the scientific and therapeutic fields. The high numbers of New Psychoactive Substances (NPS) that appears every year in the illicit drug market is an extremely dynamic phenomenon that has modified the landscape of users behavior. The exchange of cheap, easily available and replaceable substances contributes to this great dynamism and constant change of the market. The NPS consumption pose a potential burden on safety and public health. In fact, the concurrent intake of NSP with other drugs (alcohol, other classic drugs of abuse or NPS) associated with their difficult identification and characterization, make the interventions in the ER mainly symptomatic. Therefore, the knowledge of their mechanisms of action and the possible clinical complications resulting from their intake, represent a starting point to counter the risks related to the circulation and consumption of these substances.

Rev2Q6: A Limitations and strengths section is also needed.

AA: As suggested by Reviewer 2 in the conclusions we stated that: “Given the great potential toxicological and forensic value of epigenetic changes induced by exposure to drugs of abuse, the overall strength of the present narrative review is the suggestion of a translational evaluation of the pharmaco-toxicological effects of NPS widely reported by preclinical and clinical literature. However, this aspect also represents a weakness, since the great variety of environmental and non-environmental factors that can influence epigenomic”.
